# Tracking the Emergence and Dissemination of a $bla_{NDM-23}$ Gene in a Multidrug Resistance Plasmid of *Klebsiella pneumoniae*

Neris García-González,[a,b] Beatriz Beamud,[a,b] Begoña Fuster,[c] Salvador Giner,[d] Maria Victoria Domínguez,[e] Antonia Sánchez,[f] Jordi Sevilla,[a] Teresa M. Coque,[g,i] Concepción Gimeno,[c] Fernando González-Candelas,[a,b,h] on behalf of the Networked Laboratory for Antimicrobial Resistance Surveillance of Comunitat Valenciana (Spain)

[a]Institute for Integrative Systems Biology (CSIC-UV), Valencia, Spain

[b]Joint Research Unit Infection and Public Health FISABIO-University of Valencia, Institute for Integrative Systems Biology (UV-CSIC), Valencia, Spain

[c]Microbiology Service, General University Hospital, Valencia, Spain

[d]Microbiology Service, Hospital La Fe, Valencia, Spain

[e]Microbiology Service, Hospital Arnau de Vilanova, Valencia, Spain

[f]Microbiology Service, General University Hospital, Alicante, Spain

[g]Department of Microbiology, Ramón y Cajal University Hospital and Ramón y Cajal Health Research Institute (IRYCIS), Madrid, Spain

[h]Network Research Centre for Epidemiology and Public Health, Madrid, Spain

[i]Network Research Centre for Infectious Diseases (CIBERINFEC), Instituto de Salud Carlos III, Madrid, Spain

**ABSTRACT** Since the discovery of $bla_{NDM-1}$, NDM $\beta$-lactamases have become one of the most widespread carbapenemases worldwide. To date, 43 different NDM variants have been reported but some, such as $bla_{NDM-23}$, have not been characterized in detail yet. Here, we describe the emergence of a novel $bla_{NDM-23}$ allele from a $bla_{NDM-1}$ ancestor and the multidrug resistance plasmid that has disseminated it through a *Klebsiella pneumoniae* ST437 clone in several Spanish hospitals. Between 2016 and 2019, 1,972 isolates were collected in an epidemiological survey for extended-spectrum-$\beta$-lactamase (ESBL)-producing *Klebsiella pneumoniae* in the Comunitat Valenciana (Spain). Three carbapenem-resistant strains failed to be detected by carbapenemase-producing *Enterobacteriaceae* (CPE) screening tests. These isolates carried a $bla_{NDM-23}$ gene. To characterize this gene, its emergence, and its dissemination, we performed antimicrobial susceptibility tests, hybrid sequencing with Illumina and Nanopore technologies, and phylogenetic analyses. The MICs of the $bla_{NDM-23}$ allele were identical to those of the $bla_{NDM-1}$ allele. The $bla_{NDM-23}$ allele was found in 14 isolates on a 97-kb nonmobilizable, multidrug-resistant plasmid carrying 19 resistance genes for 9 different antimicrobial families. In this plasmid, the $bla_{NDM-23}$ gene is in the variable region of a complex class 1 integron with a singular genetic environment. The small genetic distance between $bla_{NDM-23}$-producing isolates reflects a 5-year-long clonal dispersion involving several hospitals and interregional spread. We have characterized the genomic and epidemiological contexts in the emergence and community spread of a new $bla_{NDM-23}$ allele in a multidrug resistance (MDR) plasmid of *Klebsiella pneumoniae*.

**IMPORTANCE** At a time when antimicrobial resistance has become one of the biggest concerns worldwide, the emergence of novel alleles and extremely drug-resistant plasmids is a threat to public health worldwide, especially when they produce carbapenem resistance in one of the most problematic pathogens, such as *Klebsiella pneumoniae*. We used genomic epidemiology to describe the emergence of a novel NDM-23 allele and identify it in a MDR plasmid that has disseminated through a *K. pneumoniae* ST437 clone in several hospitals in Spain. Using bioinformatic and phylogenetic analyses, we have traced the evolutionary and epidemiological route of the new allele, the hosting plasmid, and the strain that carried both of them from Pakistan to Spain. A better understanding of the NDM-producing *K. pneumoniae* populations and plasmids has made evident the spread of this clone

Address correspondence to Neris García-González, neris.garcia@uv.es, or Fernando González-Candelas, fernando.gonzalez@uv.es.

The authors declare no conflict of interest.

through the region, enhancing the importance of genomic surveillance in the control of antimicrobial resistance.

**KEYWORDS** *Klebsiella pneumoniae*, novel NDM, complex class 1 integron, XDR plasmid, genomic epidemiology, MDR plasmid

The global dissemination of carbapenem-resistant *Enterobacteriaceae* has become a major threat to public health. Since the discovery of $bla_{NDM-1}$ in a *Klebsiella pneumoniae* strain in 2008 (1), NDM $\beta$-lactamases have become one of the most widespread carbapenemases worldwide (2), due to their rapid evolution and dissemination via multidrug resistance (MDR) plasmids (3). $bla_{NDM}$ genes are predominantly found in the family *Enterobacteriaceae*, in which *K. pneumoniae* is the species with the highest frequency of these genes (4). Indeed, carbapenem-resistant *K. pneumoniae* represents the fastest-growing antibiotic resistance threat in Europe, in terms of the number of infections and mortality (5). To date, 43 different NDM variants have been reported, some of them showing an enhanced carbapenemase activity. Nevertheless, the dissemination, genetic context, and carbapenemase activity of some variants remain unclear (6).

The first case of a $bla_{NDM}$-producing *K. pneumoniae* strain in Spain was reported in 2012 (7). Since then, and despite recent increases in prevalence, the total number of cases has remained low, and the organism has mainly been associated with a few sporadic cases or small outbreaks. These outbreaks are primarily related to alleles $bla_{NDM-1}$ and $bla_{NDM-7}$ carried by IncR, IncX3, IncN2, and IncFIB plasmids and associated with ST437, ST11, ST101, and ST147 strains (8, 9).

The Consorcio Hospital General Universitario de Valencia is a health care facility complex with a reference population of almost 360,000 inhabitants in Valencia, Spain. Since 2015, after an imported case of $bla_{NDM-1}$-producing *K. pneumoniae*, the hospital noticed a substantial increase of this pathogen in the hospital (10). In 2018, an epidemiological surveillance program for *K. pneumoniae* was established in the Comunitat Valenciana (CV) region in Spain. Here, we report the emergence and dissemination of an ST437 clone carrying a $bla_{NDM-23}$ in an MDR plasmid. In our work, we observed that this lineage was present in 5 hospitals in the region for 5 years. We also give a detailed characterization of the novel $bla_{NDM-23}$ gene, its immediate genetic context, and the antimicrobial resistance phenotype it confers. A previous $bla_{NDM-23}$ allele had been deposited as a complete coding sequence (CDS) in GenBank (accession no. MH450214.1). Still, no information about its genetic context, bacterial host, or phenotypes has been published so far.

## RESULTS

**$bla_{NDM-23}$ is located in a nonmobilizable, nontypeable, multidrug resistance plasmid.** The $bla_{NDM-23}$ sequence differs from that of $bla_{NDM-1}$ in one nonsynonymous substitution at codon 101 (I101L). The MICs for TOP10 *Escherichia coli* transformants carrying these alleles were coincident for all the antibiotics (Table 1); thus, $bla_{NDM-23}$ shows the same antimicrobial susceptibility as $bla_{NDM-1}$, which means high-level resistance to carbapenems and all $\beta$-lactams (see Fig. S1 in the supplemental material). Cefiderocol demonstrated high *in vitro* activity toward $bla_{NDM-1}$ and $bla_{NDM-23}$ transformants.

The long hybrid assembly of isolate 146KP-HG revealed that the $bla_{NDM-23}$ gene resides on a 97-kb plasmid, referred to here as p146KP-NDM23. In addition to p146KP-NDM23, this isolate carried three plasmids corresponding to a 118-kb phage-like plasmid and 5-kb and 3-kb plasmids.

p146KP-NDM23 is a multidrug resistance plasmid that, in addition to $bla_{NDM-23}$, has 18 more genes associated with reduced susceptibility to nine different antimicrobial families, including $\beta$-lactams, sulfonamides, aminoglycosides, trimethoprim, fluoroquinolones, tetracycline, chloramphenicol, fosfomycin, tunicamycin, and rifampicin. All these genes are inserted in a large multiresistance region (MRR) that contains different putative transposable modules (Fig. 1A), which have been previously detected in IncFII plasmids (11, 12). They include the gene array comprising $bla_{OXA-1}$, a truncated *catB* gene, and *aac(6')-Ib-cr* flanked by two copies of IS26; the tetracycline resistance gene,

**TABLE 1** Antimicrobial susceptibility testing for the clinical isolates and the TOP10 *E. coli* transformants carrying *bla*<sub>NDM-1</sub> and *bla*<sub>NDM-23</sub> genes R, resistant; S, susceptible

| | Clinical isolates | | | | Transformants | | | |
| | 179KP-HG (NDM-1) | | 146KP-HG (NDM-23) | | TOP10-NDM-1 | | TOP10-NDM-23 | |
| Antimicrobial agent | MIC ($\mu$g/mL) | Category | MIC ($\mu$g/mL) | Category | MIC ($\mu$g/mL) | Category | MIC ($\mu$g/mL) | Category |
|---|---|---|---|---|---|---|---|---|
| Ampicillin | >8 | R | >8 | R | >8 | R | >8 | R |
| Amoxicillin-clavulanate | >32 | R | >32 | R | >32 | R | >32 | R |
| Piperacillin-tazobactam | >8 | R | >16 | R | >16 | R | >16 | R |
| Cefuroxime | >16 | R | >8 | R | >8 | R | >8 | R |
| Cefotaxime | >8 | R | >32 | R | >32 | R | >32 | R |
| Cefepime | >16 | R | >8 | R | >8 | R | >8 | R |
| Cefiderocol[a] | 0.75 | S | >256 | R | 0.5 | S | 0.75 | S |
| Aztreonam | >4 | R | >4 | R | ≤1 | S | ≤1 | S |
| Ertapenem[b] | >64 | R | >64 | R | >64 | R | >64 | R |
| Imipenem[b] | >64 | R | >64 | R | >64 | R | >64 | R |
| Meropenem[b] | >64 | R | >64 | R | >64 | R | >64 | R |
| Amikacin | ≤8 | S | ≤8 | S | ≤8 | S | ≤8 | S |
| Ciprofloxacin | >1 | R | >1 | R | ≤0.06 | S | ≤0.06 | S |
| Fosfomycin | ≤16 | S | ≤16 | S | ≤16 | S | ≤16 | S |
| Tigecycline | ≤8 | S | >2 | S | ≤1 | S | ≤1 | S |
| Trimethoprim | >4 | | >4 | | ≤2 | | ≤2 | |
| Cotrimoxazole | >4/76 | R | >4/76 | R | ≤2/38 | S | ≤2/38 | S |
| Colistin | ≤2 | S | ≤2 | S | ≤2 | S | ≤2 | S |

[a]The MIC was checked by test strips, unless otherwise indicated. R, resistant; S, susceptible.
[b]The MIC was checked by the broth microdilution method.

*tetA*, associated with the Tn*As1* transposase; genes *aac(3)-IIa* and *tmrB* linked to a copy of IS*26*; and a ΔTn*3* (*bla*<sub>TEM-1B</sub>) with the transposase (*tnpA*) disrupted by an IS*Ecp1*-*bla*<sub>CTX-M-15</sub>, with *sul2*, *strA*, and *strB* genes located downstream of *bla*<sub>TEM-1B</sub>.

The *bla*<sub>NDM-23</sub> gene is in a complex class 1 integron associated with ISCR1 (Fig. 1B), as previously described (4). In the first variable region of the integron, we found *acc (6′)-Ib-cr*, *arr3*, *drfA27*, and *aadA16* gene cassettes. Then, the ISCR1 region comprised an *aadA1* gene, a truncated *bla*<sub>OXA-10</sub> gene, and a truncated IS*Aba125* downstream of the *bla*<sub>NDM-23</sub> gene, while upstream we found a *ble*<sub>MBL</sub> gene, a phosphoribosyl anthranilate isomerase gene (*trpF*), and the oxidoreductase gene *dsbC*.

The plasmid p146KP-NDM23 type could not be identified by either replicon (Inc) or relaxase mobilization (MOB) typing. Although a BLAST search with RepA returned 24 plasmids with 100% identity to the query sequence (Table S1), no incompatibility type could be assigned to any of them. Apart from the *repA* gene, no homology was found for the rest of the sequence of p146KP-NDM23 and any of the 24 plasmids. We could not detect any relaxases or transfer genes in p146KP-NDM23. Thus, mobilization of p146KP-NDM23 could occur only in *trans* with the help of coresident plasmids; nevertheless, conjugation assays showed no mobilization to the receptor cells.

**Origin of the p143KP-NDM23 plasmid.** When comparing the p146KP-NDM23 sequence with the aforementioned databases, the most similar plasmid to p146KP-NDM23 in terms of identity and shared genes (100% coverage and identity, Table S2) was pKDO1 (accession no. JX424423). Plasmid pKDO1 was isolated from an ST416 *K. pneumoniae* strain in the Czech Republic (13). pKDO1 contains all the genes found in plasmid p146KP-NDM23 with rearrangements but not the ISCR1 complex class 1 integron, where the *bla*<sub>NDM-23</sub> gene is embedded (Fig. 2).

However, complex class 1 integrons and ISCR1 associated with *bla*<sub>NDM-23</sub> genes have been described previously (4). The structure around the *bla*<sub>NDM-23</sub> gene in p146KP-NDM23 is not the usually conserved structure found around *bla*<sub>NDM</sub> genes. In this case, we found the presence of a truncated *bla*<sub>OXA-1</sub> downstream of *bla*<sub>NDM</sub>-ΔIsaba125. This genetic environment has been described previously in different plasmids (Table S3) but carrying the *bla*<sub>NDM-1</sub> allele. An IncN3 plasmid, pLK78 (accession no. KJ440075), was found to carry the environment most similar to that of *bla*<sub>NDM-23</sub> in p146KP-NDM23.

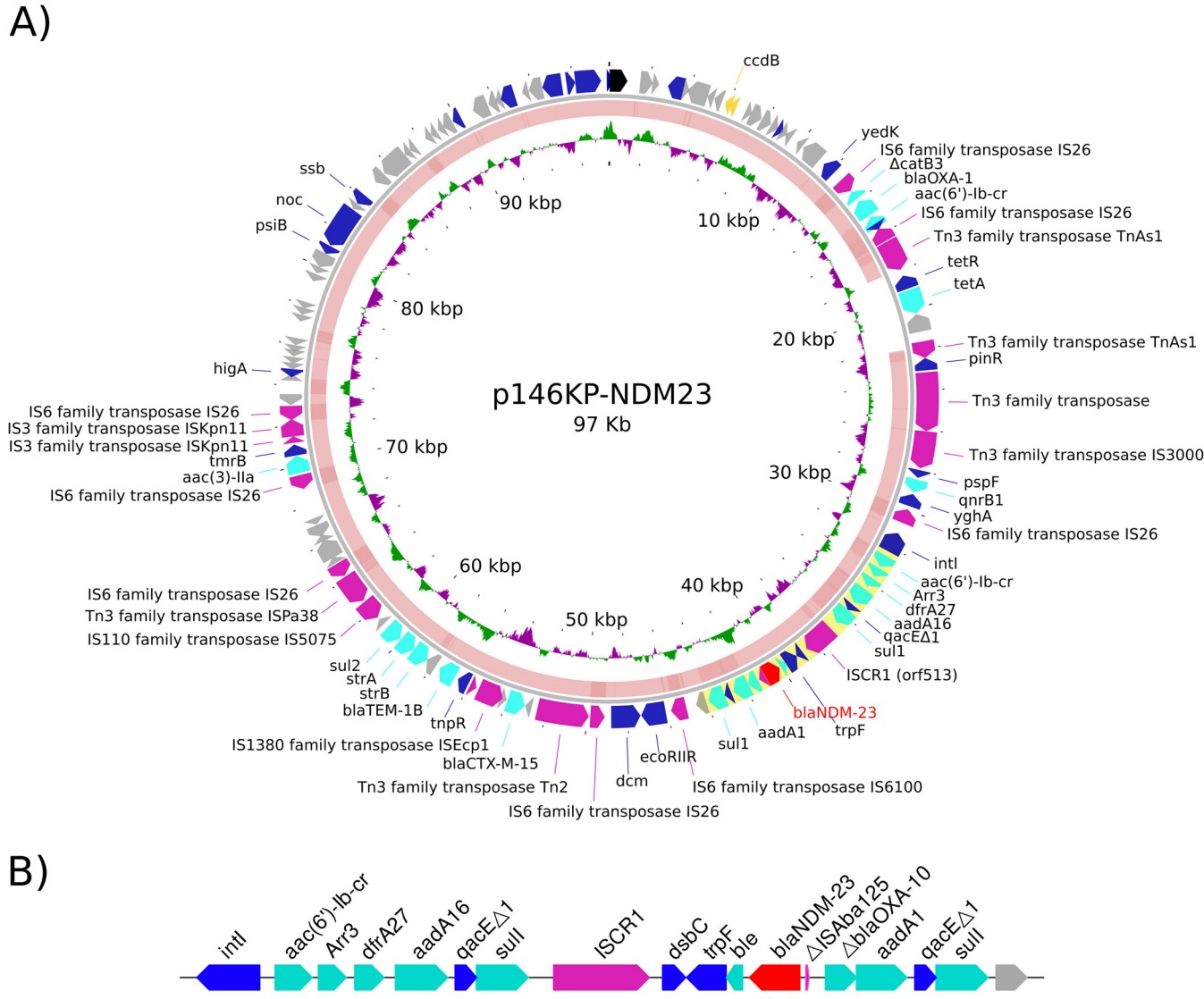

**FIG 1** (A) Gene content and structure of plasmid p146KP-NDM23. The $bla_{NDM-23}$ embedded in the second variable part of a ISCR1 complex class 1 integron is highlighted in yellow. The middle ring shows the BLAST comparison between p146KP-NDM23 and p179KP-HG. The inner ring shows the GC skew of the plasmid. (B) ISCR1 complex class 1 integron where the $bla_{NDM-23}$ gene is embedded.

Plasmid pLK78 was isolated in 2012 in Taiwan and collected from a *K. pneumoniae* strain (14). Plasmids pLK78 and p146KP-NDM23 have only the ISCR1 complex class 1 integron with the $bla_{NDM}$ gene in common (99.9% identity and 100% coverage), while the rest of the plasmid has no similarity. The identity between these two integrons is not perfect because of the single substitution that differentiates the $bla_{NDM-1}$ allele from $bla_{NDM-23}$.

We detected the presence of the p146KP-NDM23 plasmid, the ISCR1 complex class 1 integron, and the previous pKDO1 and pLK78 plasmid sequences with coverages of 95.1%, 100%, 99.83%, and 92.39%, respectively, in the genome of a *K. pneumoniae* isolate named PN4 (15). PN4 was involved in a multispecies outbreak in 2010 in Pakistan. It was not possible to decipher the structure of each mobile genetic element in this strain, because PN4 is available only as a draft genome assembly. Nonetheless, we found a possible origin where the plasmid could have formed, as we identified all the necessary structures in a single isolate.

When comparing the plasmid p146KP-NDM23 with the other strain sequence obtained from long reads in this work, 179KP-HG, we found a plasmid almost identical to p146KP-NDM23 but carrying the $bla_{NDM-1}$ allele and lacking a Tn*As1* transposable

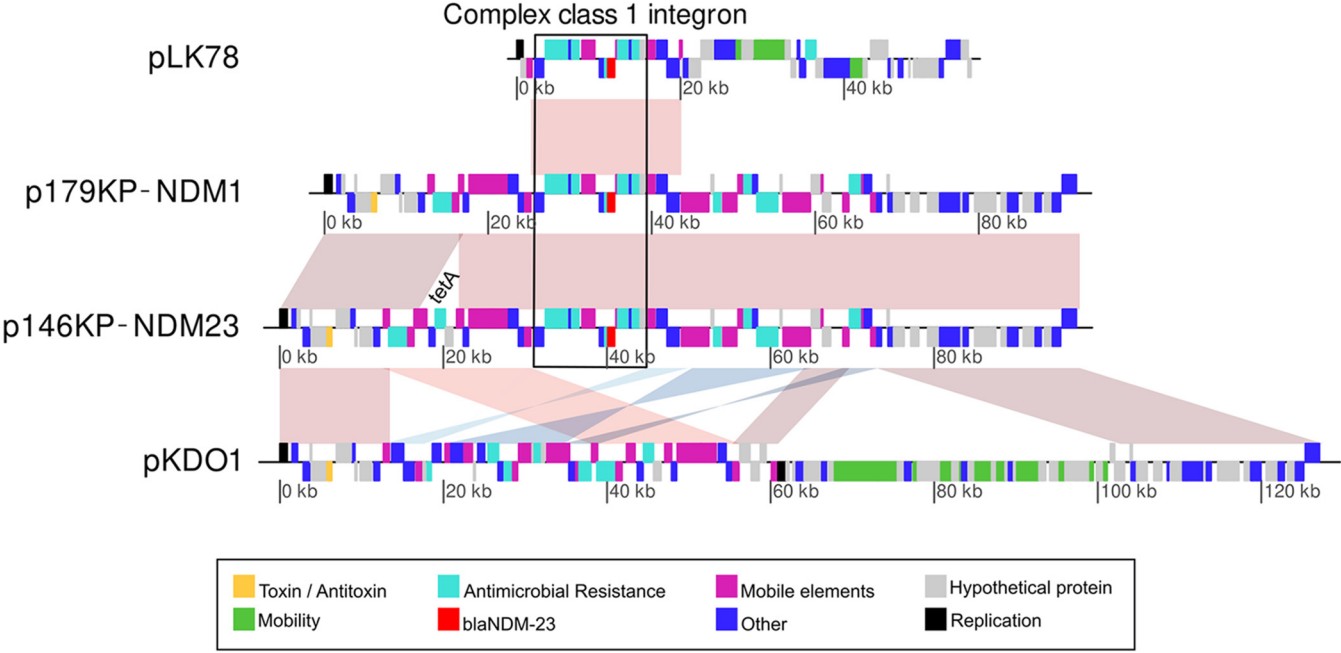

**FIG 2** Structure comparison of plasmids pLK78, p146KP-NDM23, pKP179-NDM1, and pKDO1. pLK78 shares similarity with the integron sequence found in pKP146-NDM23 and pKP179-NDM1, while pKDO1 shares similarity with the backbone of those plasmids.

element carrying a *tetA* resistance gene (Fig. 1A). We named this plasmid p179KP-NDM1.

**The recent evolutionary history of *bla*<sub>NDM-23</sub>-producing isolates is associated with ST437.** We collected 1,972 isolates under the Networked Laboratory for Surveillance of Antimicrobial Resistance (NLSAR) genomic surveillance program and determined that 47 of them (2.31%) carried $bla_{NDM}$ genes. Nonetheless, 3 of those 47 isolates yielded negative results in the carbapenemase-producing *Enterobacteriaceae* (CPE) screening tests (Table S4). All of these isolates carried a $bla_{NDM}$ gene with a novel allele, $bla_{NDM-23}$.

In total, 8 $bla_{NDM-23}$-carrying strains were found, all of them belonging to ST437 (Table S5). Thus, to analyze the epidemiological and evolutionary dynamics of the clonal background of the new $bla_{NDM-23}$ gene and its plasmid, we included all the ST437 isolates collected in the NLSAR surveillance program that carried $bla_{NDM}$ genes (8 isolates) or not (14 isolates). We also included all the ST437 genomes from RefSeq (121 isolates, accessed on 27 April 2021) (Table S6).

Reads from all the samples were mapped to the 146KP-HG closed chromosome derived from the hybrid assembly. The mapping coverage values ranged from 91.6% to 99.9% with an average of 96.27% (Table S7). Integrative mobile genetic elements were identified and removed from the final alignment: five complete prophages identified by PHASTER (Table S8) and an ICEkp yersiniabactin. This resulted in a sequence alignment of 143 complete genomes spanning 5,063,248 bp, of which 5,960 corresponded to variant positions (single nucleotide polymorphisms [SNPs]).

The phylogenetic analysis revealed that the ST437 global population collected in this work is divided into three main groups corresponding to different capsular types. Only the isolates belonging to capsular type KL36 (Fig. 3) were kept for further analysis, as all the $bla_{NDM-23}$ producers were KL36 isolates. The distance matrix for all the isolates is shown in Table S9, and the maximum-likelihood (ML) tree can be found in Fig. S2. The model used to reconstruct the tree was GTR+F+I+G4.

Within KL36, all Spanish isolates were clustered in cluster A or cluster B. $bla_{NDM-23}$ carriers were found only in cluster A (Fig. 3). This cluster contains 16 isolates, all of them with a $bla_{NDM}$ gene in the accessory genome and collected in Spain. Nine isolates were collected in the NLSAR surveillance program, while seven isolates were downloaded from the database. These database genomes were isolated in 2016 (9, 16), and

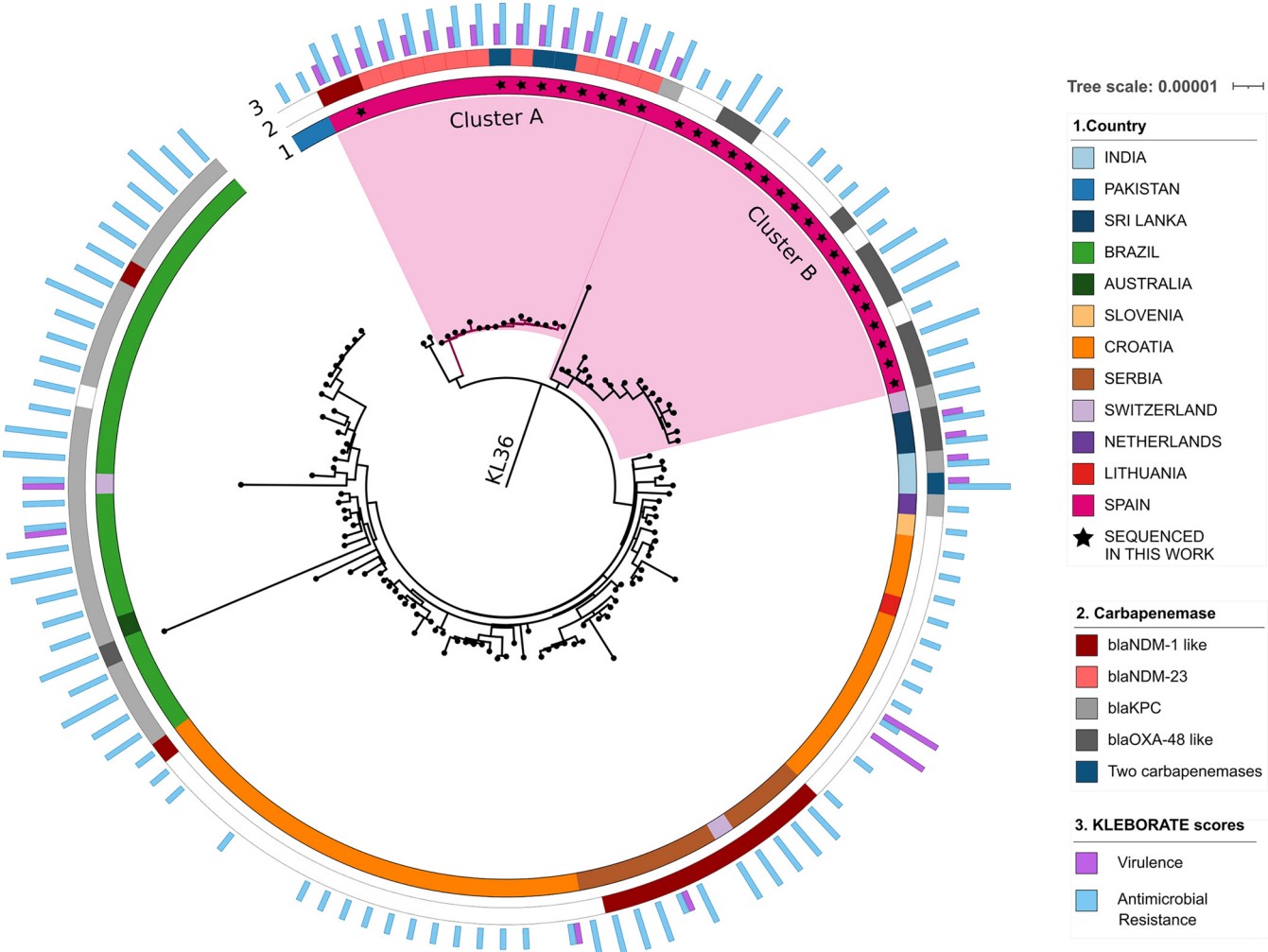

**FIG 3** Whole-genome maximum-likelihood tree of ST437 KL36 isolates. Spanish isolates are highlighted in pink. Colors in the inner ring (1) indicate the country of origin of each sample. A star inside this ring indicates that the isolate was sequenced in this work. Colors in the center ring (2) represent carbapenem-resistant genes. In the outer ring (3), blue bars show the resistance score while purple bars show the virulence score obtained from Kleborate (30). The scale represents nucleotide divergence as substitutions per site.

all of them carried the $bla_{NDM-23}$ allele. This implies that the $bla_{NDM-23}$ allele was already in Spain before we noticed it in the NLSAR surveillance. The low number of SNPs among cluster A genomes (0 to 30, with an average of 11) and the highly conserved accessory genome of this clade suggest a recent common ancestry. Temporal analysis showed a very recent most recent common ancestor (MRCA) of cluster A in mid-2013 ($R^2 = 0.47$). In addition, this clade shares a yersiniabactin virulence factor (*ybt9*) in its chromosome.

The remaining isolates collected in Spain and in the NLSAR surveillance program were grouped in cluster B. Although cluster B contains NLSAR isolates collected from the same hospitals and on the same dates as those in cluster A, the phylogenetic tree shows that these clusters likely represent two different introductions of ST437-KL36 in the region and that the MRCA of cluster B isolates occurred around 1999 ($R^2 = 0.60$). This is also supported by the average 141 SNPs (range, 84 to 275) and the very different accessory genomes between cluster A and cluster B (range, 224 to 1,281 genes). The phylogenetic temporal analysis with TempEst showed that the ancestor of clusters A and B occurred in 1996 ($R^2 = 0.35$). Hence, the Spanish non-NDM-producing ST437 isolates sequenced in this work were not the founder population of the $bla_{NDM-23}$-producing isolates.

The phylogeny (Fig. 3) shows that the closest isolates to cluster A are two samples from Pakistan collected in 2011 (accession no. GCA_900181335 and GCA_900181325).

These isolates show an average difference of 146 SNPs (range, 131 to 157) with cluster A isolates. These SNPs are dispersed over the whole genome, and they do not cluster in one or a few loci. Hence, we can eliminate the possibility that they have arisen from recombination or horizontal gene transfer. In addition, the Pakistan and cluster A samples share an *E. coli* phage, phiV10 (accession no. NC_007804), inserted in the chromosome, that is not present in any of the other samples, further supporting their common ancestry. TempEst analysis estimated that the date of the MRCA between the Pakistan and cluster A samples was in 1999 ($R^2 = 0.78$).

**Transmission of plasmid p146KP-NDM23 carrying $bla_{NDM-23}$ by recent clonal dissemination.** Clade A comprises all $bla_{NDM-23}$ isolates, and all of them carry this gene in p146KP-NDM23. All 8 $bla_{NDM-23}$-carrying strains were nonsusceptible to almost all the antimicrobials tested, except for colistin, tigecycline, and, occasionally, amikacin and fosfomycin (Table S10). All the other antimicrobial resistance determinants found in these strains were carried in the p146KP-NDM23 plasmid. No other antimicrobial resistance genes or relevant mutations (*ompK* and *parC*) were found in the chromosome or other plasmids of these strains except for intrinsic resistance genes ($bla_{SHV-11}$, *fosA*, *oqxA*, and *oqxB*). Additionally, we found both $bla_{NDM-23}$ and $bla_{OXA-48}$ genes in 3 isolates.

Remarkably, when cefiderocol activity against isolates 179KP-HG ($bla_{NDM-1}$) and 146KP-HG ($bla_{NDM-23}$) was tested, there was a big difference in the MICs, which were 0.7 and >256 $\mu$g/mL, respectively (Table 1). This difference in the MIC can be explained by the fact that 179KP-HG lacks the *baeSR* regulon while 146KP-HG and the remaining isolates of the clade carry the regulon with a mutation in the *baeS* gene (F3L), recently related to reduced susceptibility to cefiderocol (17). On the basis of the minimal number of differences found between the chromosomes of clade A isolates, the identical accessory genome, and the date of the MRCA of the clade (mid-2013), we assumed that clade A represents a case of recent clonal dissemination. In addition to hospital H0 in Madrid (accession no. GCA_011684095) and hospital H1 in Valencia, this dissemination also affected at least three different hospitals in the Comunitat Valenciana, spanning from 2016 until at least 2019, when our sampling concluded. This clade comprises two basal isolates (accession no. GCA_011684095 and isolate 179KP-HG) carrying the carbapenemase gene $bla_{NDM-1}$ and 14 isolates that carry the novel $bla_{NDM-23}$ allele. The index case of the dissemination of this clone, which yielded the isolate with accession no. GCA_011684095, is the most basal genome of cluster A and was also the index case of a multiclonal and multispecies outbreak detected in hospital H0 in Madrid (16). This patient was a 36-year-old man who was a native of Pakistan with residence in Valencia. The patient had received health care assistance in Pakistan following a traffic accident in November 2014. On 26 August 2015, upon return from Pakistan, he was admitted to the neurosurgery ward in H0 in Madrid. In December 2015, the patient was transferred to Valencia and admitted to hospital H1. A few months later, this patient went through the emergency ward in H2 and was admitted to the general ward. This explains the fast spread of the lineage through the region but also points to Pakistan as the possible origin of the strain causing this dissemination event.

The $bla_{NDM-23}$ mutation had to occur between the arrival of the clones at the hospital and the first detection of $bla_{NDM-23}$ genes in the hospital. The first clone of clade A carried $bla_{NDM-1}$. After that, the p146KP-NDM23 plasmid remained genetically stable over time. We found 100% identity for p146KP-NDM23 plasmid among all the isolates but with an average coverage of 97.5%, ranging from 84% to 100% (Table S11; Fig. 4). The plasmid has accrued several structural changes, mainly deletions (Fig. 4). The accessory genome of the corresponding samples did change during the dissemination process, with gains and losses of complete mobile genetic elements. For instance, samples 262KP-HG, HAV1_09, and KP-HGUA01_12 have an additional IncL plasmid harboring a $bla_{OXA-48}$ gene. Contrarily, some samples have lost a few complete plasmids: sample KP-HGUA01_12 lacks 118-kb, 5-kb, and 3-kb plasmids, and samples HLF2_50, HGV2_363, and HGV2_364 lack plasmid 118Kb.

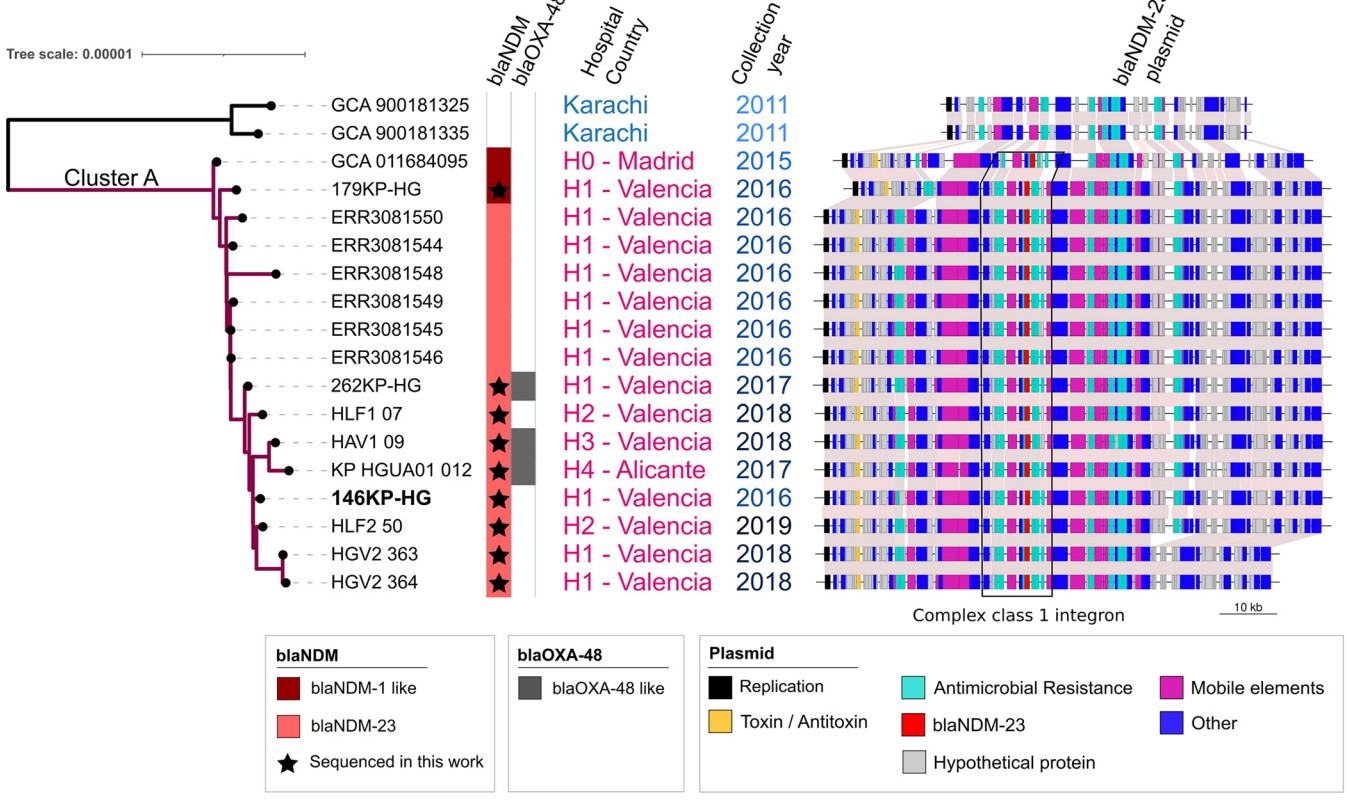

**FIG 4** Dissemination of the ST437 clone carrying the $bla_{NDM23}$ gene involves 5 different hospitals over 5 years. Isolates sequenced in this work are marked with a star inside the $bla_{NDM}$ strip. For each isolate, the structural variants of plasmid p146KP-NDM23 are shown.

## DISCUSSION

The dissemination of $bla_{NDM}$ genes in a broad range of Gram-negative bacteria, such as *K. pneumoniae*, via MDR plasmids has established NDM as a global major public health threat (3). The evolutionary and molecular mechanisms acting on plasmids and the genes they contain are well understood at a general scale, but there is little detailed information on how they operate at short time scales (18). A better understanding of these processes is thus essential for controlling and limiting the spread of antimicrobial resistance genes (ARGs).

Here, we describe the emergence and dissemination of a multidrug-resistant plasmid associated with the clonal dissemination of a *K. pneumoniae* ST437 strain carrying a new $bla_{NDM-23}$ carbapenemase gene and 18 more antimicrobial resistance genes. We have also determined when and where a point mutation in the plasmid sequence produced a change from a $bla_{NDM-1}$ allele to $bla_{NDM-23}$ during the dissemination of this clone.

The multidrug resistance plasmids in which the $bla_{NDM-1}$ and $bla_{NDM-23}$ genes were identified were named pKP179-NDM1 and pKP146-NDM23, respectively. These plasmids contain the $bla_{NDM}$ alleles in a complex class 1 integron associated with ISCR1, in addition to 18 genes that confer resistance to nine different antimicrobial families. These plasmids were found to be very similar to plasmid pKDO1, described in the Czech Republic in 2009 (13), while the complex class 1 integron was identified in plasmid pLK78, isolated in Taiwan in 2012 (14). Contigs from the *K. pneumoniae* strain PN4 genome, isolated in Pakistan in 2012, included all the plasmids (pKP146-NDM23, pLK78, and pKDO1). Hence, we hypothesize that pKP146-NDM23 originated in this lineage. We found a conjugation module in pKDO1 but not in pKP179-NDM1 or pKP146-NDM23, suggesting that the latter plasmids lost their mobilization capabilities before their mobilization to the ancestor of clade A.

Although conjugation assays for p146KP-NDM23 were negative, our bioinformatic analyses show the presence of the complex class 1 integron in different plasmids (Table S6).

This might be explained either because the integron is flanked by IS26 transposases, which could mediate its transposition, or, although it has not been proven experimentally, by the independent mobilization of ISCR1 elements (19). Nevertheless, further work will be needed to understand the mobilization of these genetic elements in different strains and species.

The close similarity of the chromosome sequences of isolates in clade A (Fig. 4) and two isolates from Pakistan, the origin of the p146KP-NDM23, and the epidemiological link between the index case in Spain and that country strongly suggest that the origin of cluster A sequences is Pakistan and not in other ST437 isolates previously found in Spain (cluster B). Moreover, the results indicated a recent emergence and clonal expansion of clade A in the Comunitat Valenciana.

Whereas the outbreak that occurred in 2015 at H0 in Madrid was contained at a single hospital, the spread of this clone throughout the Comunitat Valenciana in the following years, until at least 2019, represents the first case of interhospital and community spread of this clone in our country. In some of these hospitals, including H0 (16), the $bla_{NDM-23}$ allele was not detectable by some carbapenemase tests, and its actual spread might be underestimated. Remarkably, during its recent and short (5-year) dissemination, the plasmid p146KP-NDM23 has shown complete conservation at the nucleotide level but several differences at the structural level.

Worrisomely, the *K. pneumoniae* ST437 clone spreading in the region is susceptible to only a few second-line antimicrobials with a risk of toxicity or other safety concerns (20). Furthermore, a high resistance level to the new cephalosporin cefiderocol was detected. Although little is known about the mechanisms of resistance to this antimicrobial, resistance has been reported for *K. pneumoniae* with mutations in *cirA* genes when combined with NDM production (21) or the combination of $\beta$-lactamases with mutations in the *baeS* gene (*baeSR* regulon) (17, 22). In our study, NDM-TOP10 transformants did not show cefiderocol-hydrolyzing activity, meaning that neither NDM-1 or NDM-23 hydrolyzed cefiderocol by itself. However, when tested in clinical samples, we observed that 146KP-HG showed a high resistance level (MIC > 256 $\mu$g/mL). This isolate, as the ones in clade A, carries a combination of a mutation in the *baeS* gene and other antimicrobial resistance (AMR) determinants already reported as being related to cefiderocol resistance. However, their MIC was lower (4 versus 256 $\mu$g/mL) (17). Moreover, the clone carries a yersiniabactin virulence factor in the chromosome, which encodes an iron-scavenging molecule that enhances its capacity to cause disease associated with bacteremia and tissue-invasive infections (23). The combination of the limited number of effective therapeutic options available to treat infections caused by this clone and the increasing number of infections in this region make this clone and its plasmid of particular concern for local and regional public health.

An important goal of our surveillance program is to inform about new threats and facilitate improved intervention strategies for controlling their spread. Infectious disease specialists should be aware of the possibility of finding this clone in their hospitals. Remarkably, carbapenemase detection was negative when some of these patients were tested for colonization with carbapenemase-producing organisms (16). This highlights the importance of updating diagnostic techniques in hospitals to prevent further dissemination of this clone and detect this carbapenemase. As a nonmobilizable plasmid, the plasmid identified here can be transmitted only by clonal dissemination, and hospital control measures will have to focus on timely detection of $bla_{NDM-23}$ and on finding the source of this clone and removing it from hospitals.

## MATERIALS AND METHODS

**Setting, sample collection, and bacterial isolation.** During the period 2016 to 2019, we conducted the NLSAR study of ESBL or carbapenemase-producing *K. pneumoniae* in the Comunitat Valenciana (Spain). Eight hospitals in the region participate in the Networked Laboratory. These are reference hospitals for more than 25 hospitals and health care centers. The first 30 clinical isolates of the month (if they had that many) identified as ESBL- or carbapenemase-producing *K. pneumoniae* in each hospital were included in the study. Additionally, 10 susceptible controls for each month and historical and

retrospective samples from 2004 to 2017 were included. For epidemiological studies, relevant clinical data, including isolate collection date, hospital ward, gender, and age, were obtained from hospital records.

**Antimicrobial susceptibility and CPE screening test.** Antimicrobial susceptibility tests were performed by the broth microdilution method using the MicroScan WalkAway (Beckman Coulter) automated system. The tested antibiotics varied depending on the hospital where each sample was collected. Susceptibility breakpoints were interpreted according to the recommendations of the EUCAST 2021 guidelines (https://www.eucast.org/clinical_breakpoints/). Carbapenemase production was confirmed upon admission to the hospital using different methods. Cefiderocol was tested using MIC test strips (Liofilchem MTS).

**Transformation and conjugation assays.** To evaluate the antimicrobial susceptibility of the relevant $bla_{NDM}$-carrying isolates, we performed cloning and transformation assays. The complete sequences of $bla_{NDM}$ genes were amplified as previously described (24). The PCR fragments were cloned using the TOPO TA cloning kit (Life Technologies) and transformed into the *E. coli* TOP10 background (Life Technologies). Transformants carrying the $bla_{NDM}$ genes were selected on plates containing kanamycin (50 $\mu$g/mL) and confirmed by PCR as described above. Antimicrobial susceptibility testing of the transformants was performed using the MicroScan WalkAway (Beckman Coulter) automated system. For carbapenems, the MIC was checked by the broth microdilution method. Cefiderocol was tested using MIC test strips (Liofilchem MTS). Three independent replicates for each isolate were obtained. Growth curves were constructed using ggplot2 (25).

To study the mobility of the plasmid carrying the $bla_{NDM}$ gene, conjugation assays were performed using the clinical strains encoding carbapenemases as donors and a plasmid-free azide-resistant *E. coli* J53 strain as the recipient. Three *K. pneumoniae* donors were studied: isolate 179KP-HG, carrying $bla_{NDM-1}$; isolate 146KP-HG, carrying $bla_{NDM-23}$; and isolate 262KP-HG, carrying $bla_{NDM-23}$ and $bla_{OXA-48}$. Donor and recipient cells were incubated separately overnight at 37°C with moderate shaking (150 rpm) in 1 mL of Luria-Bertani (LB) broth. Overnight cultures were 10-fold diluted and combined in a 1:1 donor-to-recipient ratio. Precipitated conjugation mixtures were cultured overnight at 37°C in a nitrocellulose membrane (0.22-$\mu$m pore diameter). Finally, recipients, donors, and transconjugants were selected on LB agar containing 0.5 $\mu$g/mL of ertapenem, 100 $\mu$g/mL of sodium azide, or both. Technical and biological replicates for each isolate were performed. Mobilization of the carbapenemase genes into *E. coli* J53 was confirmed by CHROMagar and PCR. PCRs for the $bla_{OXA-48}$ gene were performed as previously described (26).

**Whole-genome sequencing and comparative analyses.** Sequencing was performed in an Illumina NextSeq 500 platform using the Nextera XT library preparation kit. One isolate carrying $bla_{NDM-23}$ (146KP-HG) and another carrying $bla_{NDM-1}$ (179KP-HG) were also sequenced using Oxford Nanopore MinION technology.

Quality and filtering of short reads were assessed using FastQC (https://www.bioinformatics.babraham.ac.uk/projects/fastqc/) and Prinseq-lite (https://prinseq.sourceforge.net/). Reads with a mean quality below 25 or a read length shorter than 60 bp or longer than 500 bp and optical duplicates were removed. Also, the last three positions in the 3′ end and positions with a quality lower than 20 were trimmed.

To study the origin and evolution of $bla_{NDM-23}$-carrying plasmids and their association with ST437 isolates, we analyzed the genetic context of all $bla_{NDM}$ genes found in the NLSAR collection. Only isolates with a genetic context similar to that of the $bla_{NDM-23}$ gene were analyzed in detail and kept for further results. To trace the evolutionary origin of these strains, we performed a comparative analysis including all ST437 genome sequences publicly available at GenBank (as of 13 June 2020). Genomes with too many contigs (>500) were excluded. In addition, we included reads of ST437 isolates deposited at ENA from a previous study (9) on the emergence of NDM-producing *K. pneumoniae* and *Escherichia coli* in Spain. These reads were processed following the workflow used for the reads obtained in our study.

Samples were *de novo* assembled using Unicycler (27), and gene annotation was performed with Prokka (28). QUAST (29) was used to assess the quality of all the assemblies. Detailed characterization was performed using Kleborate (30) and Kaptive (31). Additionally, ResFinder (32), PlasmidFinder (33), PLSDB (34), staramr (https://github.com/phac-nml/staramr), and BLAST were used to corroborate the antimicrobial resistance genes and plasmids detected. To characterize mechanisms contributing to resistance to cefiderocol, isolates were compared with the *K. pneumoniae* ATCC 13883 reference genome to identify gene mutations of previously proposed cefiderocol resistance targets (17). PHASTER (35) was used to analyze the presence of phages. To investigate the similarity between plasmids, BLAST (36), Prokka (28), and the R library genoplotR (37) were used. Plasmid relaxases, mobilization, and conjugation systems were searched using OriTFinder (38). Plasmid figures were drawn using the CGview server (39) and the genoplotR library.

Chromosome comparisons were made using Snippy (https://github.com/tseemann/snippy). For this, the 146KP-HG hybrid assembled chromosome was used as the reference. Minimum quality, minimum site depth for calling alleles, and minimum proportion for variant evidence were set to 60, 5, and 0.75, respectively. To avoid including highly divergent regions in the analyses, five complete prophages identified by PHASTER and an ICEkp yersiniabactin inserted into the reference chromosome were removed from the genome alignment. The quality of the genome sequence alignment was assessed using AMAS.py (40). We used snp-sites (41) to extract SNP positions from the whole-genome alignment. Distance matrices were obtained using snp-dists (https://github.com/tseemann/snp-dists). Reference genome mapping coverage was calculated using the genomecov function of BEDTools (42). An ML phylogenetic tree was inferred from the genome alignment with IQTREE2 (43) with the TEST option to find the best-fitting substitution model (44). Ultrafast bootstrap branch supports were assessed employing 1,000 replicates (45). The ML tree was visualized using iTOL (https://itol.embl.de/). Temporal phylogenetic signal and estimation of the time to the most common recent ancestor were made using TempEst (46).

To study the shared accessory genome, unmapped reads against the 146KP-HG chromosome were mapped against the plasmids found in this sample, as described above. The remaining unmapped reads were assembled using Unicycler, and their gene content was analyzed using PROKKA and QUAST.

**Data availability.** Sequencing reads and assemblies generated in this work are available at the European Nucleotide Archive (ENA), project no. PRJEB37504 (ERS6077430 to ERS6077456).

## SUPPLEMENTAL MATERIAL

Supplemental material is available online only.
**SUPPLEMENTAL FILE 1**, PDF file, 0.3 MB.
**SUPPLEMENTAL FILE 2**, XLSX file, 0.5 MB.

## ACKNOWLEDGMENTS

We are thankful to AM Wailan for kindly providing the genome assembly of PN4.

This research was supported by project BFU2017-89594R (MICIN, Spanish Government) and by the European Union through the Operational Program of European Regional Development Fund (ERDF) of Valencia Region (Spain) 2014-2020. Additional support was provided by the Network Research Centre for Epidemiology and Public Health (CIBERESP).

Members of the Networked Laboratory for Surveillance of Antimicrobial Resistance of the Valencian Community are as follows: Neris García-González and Fernando González-Candelas, Joint Research Unit Infection and Public Health FISABIO-University of Valencia, Institute for Integrative Systems Biology, CIBER in Epidemiology and Public Health, Valencia, Spain; Begoña Fuster, Nuria Tormo, Carme Salvador, and Concepción Gimeno, Microbiology Service, Consorcio Hospital General Universitario, Valencia, Spain; Victoria Domínguez, Microbiology Service, Hospital Arnau de Vilanova, Valencia, Spain; Salvador Giner, Microbiology Service, Hospital Universitario y Politécnico La Fe, Valencia, Spain; Javier Colomina and David Navarro, Microbiology Service, Hospital Clínico Universitario, Valencia, Spain; Llúcia Martínez, Sequencing Platform, FISABIO, Valencia, Spain; Antonio Burgos and Olalla Martínez, Microbiology Section, Hospital La Ribera, Alzira, Spain; Barbara Gomila-Sard and Rosario Moreno, Microbiology Service, Hospital General Universitario, Castelló, Spain; Inmaculada Vidal, Antonia Sánchez, and Juan Carlos Rodriguez, Microbiology Service, Hospital General Universitario, Alicante, Spain; Victoria Sánchez-Hellín, Microbiology Service, Hospital General Universitario, Elx, Spain.

None of the authors declares any conflict of interest.

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
