## [Reviewer comments · Microbiology Spectrum]

Microbiology Spectrum

Tracking the emergence and dissemination of a *bla*_{NDM-23} Gene in a Multi-Drug Resistance Plasmid of *Klebsiella pneumoniae*

Neris García-González, Beatriz Beamud, Begoña Fuster-Escrivá, Salvador Giner, M^a VICTORIA DOMINGUEZ MARQUEZ, Antonia Sánchez, Jordi Sevilla, Teresa Coque, Concepción Gimeno, and Fernando Gonzalez-Candelas

Corresponding Author(s): Neris García-González and Fernando Gonzalez-Candelas, Universidad de Valencia

Review Timeline:

Submission Date:	July 6, 2022
Editorial Decision:	October 3, 2022
Revision Received:	October 21, 2022
Accepted:	January 12, 2023

Editor: Mariagrazia Perilli

Reviewer(s): Disclosure of reviewer identity is with reference to reviewer comments included in decision letter(s). The following individuals involved in review of your submission have agreed to reveal their identity: Asad U Khan (Reviewer #1)

Transaction Report:

DOI: <https://doi.org/10.1128/spectrum.02585-22>

October 3, 2022

Prof. Fernando Gonzalez-Candelas
Universidad de Valencia
Instituto de Biología Integrativa de Sistemas, I2SysBio (CSIC-UV)
Catedrático Jose Beltrán 2
Paterna, Valencia 46980
Spain

Re: Spectrum02585-22 (Tracking the emergence and dissemination of a *bla*_{NDM-23} Gene in a Multi-Drug Resistance Plasmid of *Klebsiella pneumoniae*)

Dear Prof. Fernando Gonzalez-Candelas:

Link Not Available

Sincerely,

Mariagrazia Perilli

Journals Department
Reviewer comments:

Reviewer #1 (Comments for the Author):

Review report on Tracking the emergence and dissemination of a *bla*_{NDM-23} Gene in a Multi-Drug Resistance Plasmid of *Klebsiella pneumoniae*

In this paper Tracking the emergence and dissemination of a *bla*_{NDM-23} Gene in a Multi-Drug Resistance Plasmid of *Klebsiella pneumoniae* focuses on the isolates which carries *bla*_{NDM-23} gene and for its characterization they have performed AST, Hybrid Sequencing with Illumina and Nanopore Technologies and Phylogenetic Analysis.

They have collected 1972 isolates under the NLSAR genomic surveillance program and detected 47 (2.31%) carrying NDM

genes. Nevertheless, 3 of those 47 isolates yielded negative results in the CPE screening tests. All of these isolates carried a blaNDM gene with a novel variant, blaNDM-23. In total, 8 blaNDM-23-carrying strains were found, all of them belonging to ST437. These strains were nonsusceptible to almost all the antimicrobials tested, except for colistin, tigecycline, and, occasionally, amikacin and fosfomycin. Two ST437 isolates were sequenced by ONT, one carrying blaNDM-23 (146KP-HG) and blaNDM-1 (179KP-HG) the other. Although conjugation assays for p146KP-NDM23 were negative, our bioinformatic analyses show the presence of the complex class 1 integron in different plasmids.

In this paper author describe the emergence of a novel blaNDM-23 allele from a blaNDM-1 ancestor of a multi-drug-resistant plasmid associated with clonal dissemination of a *K. pneumoniae* ST437 strain carrying a new blaNDM-23 carbapenemase gene and 18 more antimicrobial resistance genes. They have also detected when and where a point mutation in the plasmid sequence produced a change from a blaNDM-1 allele to blaNDM-23.

The MICs of blaNDM-23 were identical to those of the blaNDM-1. The blaNDM-23 variant was found in 14 isolates in a 97kb non-mobilizable multidrug-resistant plasmid carrying 19 resistance genes for 9 different antimicrobial families. In this plasmid, the blaNDM-23 gene is located in the variable region of a complex class-1 integron with a singular genetic environment. The short genetic distance between blaNDM-23 producing isolates reflects a 5-year-long clonal dispersion involving several hospitals and interregional spread.

The phylogenetic analysis revealed that the ST437 global population collected in this work is divided into three main groups corresponding to different capsular types. Only the isolates belonging to capsular type KL36 were kept for further analysis as all the blaNDM-23-producers

Try to revise language as per the journal's standard

Reviewer #2 (Comments for the Author):

Neris García-González and colleagues present a very well written and interesting manuscript dealing with the emergence and dissemination of the blaNDM-23 beta-lactamase in *K. pneumoniae*.

Just some minor comments:

1. Please, add line numbering. It is very hard for referees to indicate changes in a manuscript without line numbers. Take this into account for the revision a further manuscript.

2. Beta-lactamase should be changed to β -lactamase throughout

3. Page 1 paragraph 1:

"to describe the emergence of a novel NDM-23 allele"

Please use the term allele to refer to genes (blaNDM-23) and variants to refer β -lactamase proteins (such as in this case; e.g., NDM-23)

4. Introduction. Page 3. Paragraph 3

"We also give a detailed characterization of the novel blaNDM-23 gene, its antimicrobial properties, and its immediate genetic context. A previous variant blaNDM-23 had been deposited as a complete CDS sequence in GenBank (accession MH450214.1). Still, no information about its genetic context, bacterial host, or antimicrobial activity has been published so far."

Genes do not have antimicrobial properties by themselves. B-lactamase protein products confer a particular antimicrobial susceptibility profile. Please rephrase. Also saying that β -lactamases have antimicrobial activity sounds weird. Antibiotics have antimicrobial activity. Please also check this.

5. Introduction. Page 3. Paragraph 1

To date, 29 different NDM alleles have been reported, some of them showing an enhanced carbapenemase activity. Nevertheless, the dissemination, genetic context, and carbapenemase activity of some alleles remain unclear(6).

To date, 43 different blaNDM alleles

6. Page 7 Results

NDM-like enzymes have been noted to be associated with reduced susceptibility to cefiderocol. Please evaluate its activity against isolates present in Table 1 and comment about this in results and discussion

Also the authors comment: "The MIC values for isolates carrying these variants were coincident for all the antibiotics, thus presenting the same antimicrobial susceptibility as blaNDM-1, which means low susceptibility to carbapenems and all beta-lactams (Table 1 and figure S1)."

NDM-like enzymes do not confer low susceptibility. They confer high-level resistance to beta-lactams. Please change this.

Table 1.

I have never heard before about the "TOP10 E. fingers crossed coli" strain

Please change this

Staff Comments:

Preparing Revision Guidelines

Please return the manuscript within 60 days; if you cannot complete the modification within this time period, please contact me. If you do not wish to modify the manuscript and prefer to submit it to another journal, please notify me of your decision immediately so that the manuscript may be formally withdrawn from consideration by Microbiology Spectrum.

Response to Reviewers

Reviewer #1 (Comments for the Author):

Review report on Tracking the emergence and dissemination of a blaNDM-23 Gene in a Multi-Drug Resistance Plasmid of *Klebsiella pneumoniae*

In this paper Tracking the emergence and dissemination of a blaNDM-23 Gene in a Multi-Drug Resistance Plasmid of *Klebsiella pneumoniae* focuses on the isolates which carries blaNDM-23 gene and for its characterization they have performed AST, Hybrid Sequencing with Illumina and Nanopore Technologies and Phylogenetic Analysis.

They have collected 1972 isolates under the NLSAR genomic surveillance program and detected 47 (2.31%) carrying NDM genes. Nevertheless, 3 of those 47 isolates yielded negative results in the CPE screening tests. All of these isolates carried a blaNDM gene with a novel variant, blaNDM-23. In total, 8 blaNDM-23-carrying strains were found, all of them belonging to ST437. These strains were nonsusceptible to almost all the antimicrobials tested, except for colistin, tigecycline, and, occasionally, amikacin and fosfomycin. Two ST437 isolates were sequenced by ONT, one carrying blaNDM-23 (146KP-HG) and blaNDM-1 (179KP-HG) the other. Although conjugation assays for p146KP-NDM23 were negative, our bioinformatic analyses show the presence of the complex class 1 integron in different plasmids.

In this paper author describe the emergence of a novel blaNDM-23 allele from a blaNDM-1 ancestor of a multi-drug-resistant plasmid associated with clonal dissemination of a *K. pneumoniae* ST437 strain carrying a new blaNDM-23 carbapenemase gene and 18 more antimicrobial resistance genes. They have also detected when and where a point mutation in the plasmid sequence produced a change from a blaNDM-1 allele to blaNDM-23.

The MICs of blaNDM-23 were identical to those of the blaNDM-1. The blaNDM-23 variant was found in 14 isolates in a 97kb non-mobilizable multidrug-resistant plasmid carrying 19 resistance genes for 9 different antimicrobial families. In this plasmid, the blaNDM-23 gene is located in the variable region of a complex class-1 integron with a singular genetic environment. The short genetic distance between blaNDM-23 producing isolates reflects a 5-year-long clonal dispersion involving several hospitals and interregional spread.

The phylogenetic analysis revealed that the ST437 global population collected in this work is divided into three main groups corresponding to different capsular types. Only the isolates belonging to capsular type KL36 were kept for further analysis as all the blaNDM-23-producers

Try to revise language as per the journal's standard

We have revised the conventions noted by ASM and used them. All taxon names and genes are in italics. Protein names are in capital letters. Specific nomenclature for

tetracycline and mobile genetics elements has been revised and fixed if an error was found. Moreover, references have been edited to the journal's standards.

Reviewer #2 (Comments for the Author):

Neris García-González and colleagues present a very well written and interesting manuscript dealing with the emergence and dissemination of the blaNDM-23 beta-lactamase in *K. pneumoniae*.

Just some minor comments:

1. Please, add line numbering. It is very hard for referees to indicate changes in a manuscript without line numbers. Take this into account for the revision a further manuscript.

Changed.

2. Beta-lactamase should be changed to β -lactamase throughout

Changed.

3. Page 1 paragraph 1:

"to describe the emergence of a novel NDM-23 allele"

Please use the term allele to refer to genes (blaNDM-23) and variants to refer β -lactamase proteins (such as in this case; e.g., NDM-23)

Revised and changed allele to refer to genes and variant to refer to proteins along the text.

4. Introduction. Page 3. Paragraph 3

"We also give a detailed characterization of the novel blaNDM-23 gene, its antimicrobial properties, and its immediate genetic context. A previous variant blaNDM-23 had been deposited as a complete CDS sequence in GenBank (accession MH450214.1). Still, no information about its genetic context, bacterial host, or antimicrobial activity has been published so far."

Genes do not have antimicrobial properties by themselves. B-lactamase protein products confer a particular antimicrobial susceptibility profile. Please rephrase. Also saying that β -lactamases have antimicrobial activity sounds weird. Antibiotics have antimicrobial activity. Please also check this.

Changed to "We also give a detailed characterization of the novel *bla*_{NDM-23} gene, ~~its antimicrobial properties~~, its immediate genetic context and the antimicrobial resistant phenotype it confers. A previous *bla*_{NDM-23} allele had been deposited as a complete CDS sequence in GenBank (accession MH450214.1). Still, no information about its genetic context, bacterial host, or ~~antimicrobial activity~~ phenotypes has been published so far."

We have also checked every point where we mentioned “antimicrobial activity” to change it according to the reviewers’ recommendations.

5. Introduction. Page 3. Paragraph 1

To date, 29 different NDM alleles have been reported, some of them showing an enhanced carbapenemase activity. Nevertheless, the dissemination, genetic context, and carbapenemase activity of some alleles remain unclear(6).

To date, 43 different blaNDM alleles

Changed.

6. Page 7 Results

NDM-like enzymes have been noted to be associated with reduced susceptibility to cefiderocol. Please evaluate its activity against isolates present in Table 1 and comment about this in results and discussion

We would like to thank the reviewer for this suggestion, as we tested cefiderocol on the isolates in Table 1 and we obtained some remarkable results. When we tested cefiderocol in the *E. coli* TOP10 transformants, NDM-1 and NDM-23 carbapenemases did not show any relevant hydrolyzing activity. However, when we tested the two clinical isolates the reviewer asked about, we saw a tremendous difference. Isolate 179KP-HG (NDM-1) showed a MIC of 0.75 while 146KP-HG (NDM-23) showed a MIC of over 256. This is quite astounding because these isolates are largely clonal, sharing 99.9% of the genes and with few SNPs between them. We found that the few genes that 146KP-HG has, in addition to those 179KP-HG we found the baeSR regulon carrying the multidrug resistance *mtdABCD* genes and the F3L mutation in the *baeS* gene. Both have been recently reported to be related to cefiderocol resistance in this pathogen. We added these results in Table 1 and in the Results and Discussion sections. However, to fit these new results into the manuscript, we rearranged and moved the first subsection in Results, thus implying a change in the order of the Supplementary tables.

Also, the authors comment: "The MIC values for isolates carrying these variants were coincident for all the antibiotics, thus presenting the same antimicrobial susceptibility as blaNDM-1, which means low susceptibility to carbapenems and all beta-lactams (Table 1 and figure S1)."

NDM-like enzymes do not confer low susceptibility. They confer high-level resistance to beta-lactams. Please change this.

Changed.

"The MIC values for isolates carrying these variants were coincident for all the antibiotics, thus presenting the same antimicrobial susceptibility as NDM-1, which means ~~low susceptibility to~~ high-level resistance to carbapenems and all β -lactams (Table 1 and figure S1)."

Table 1.

I have never heard before about the "TOP10 *E. fingers crossed coli*" strain

Please change this

Sorry for the typo, we have changed it to "TOP10 *E. coli*".

January 12, 2023

Prof. Fernando Gonzalez-Candelas
Universidad de Valencia
Instituto de Biología Integrativa de Sistemas, I2SysBio (CSIC-UV)
Catedrático Jose Beltrán 2
Paterna, Valencia 46980
Spain

Re: Spectrum02585-22R1 (Tracking the emergence and dissemination of a *bla*_{NDM-23} Gene in a Multi-Drug Resistance Plasmid of *Klebsiella pneumoniae*)

Dear Prof. Fernando Gonzalez-Candelas:

Your manuscript has been accepted, and I am forwarding it to the ASM Journals Department for publication. You will be notified when your proofs are ready to be viewed.

Sincerely,

Mariagrazia Perilli
Editor, Microbiology Spectrum
